# Cost-effectiveness of a preferred intensity exercise programme for young people with depression compared with treatment as usual: an economic evaluation alongside a clinical trial in the UK

David Turner,[1] Tim Carter,[2] Tracey Sach,[1] Boliang Guo,[3] Patrick Callaghan[2]

[1]Norwich Medical School, University of East Anglia, Norwich, UK
[2]School of Health Sciences, University of Nottingham, Nottingham, UK
[3]Division of Psychiatry and Applied Psychology, University of Nottingham, Nottingham, UK

**Correspondence to**
David Turner;
David.A.Turner@uea.ac.uk

## ABSTRACT

**Objectives** To assess the cost-effectiveness of preferred intensity exercise programme for young people with depression compared with a treatment as usual control group.

**Design** A 'within trial' cost-effectiveness and cost-utility analysis conducted alongside a randomised controlled trial. The perspective of the analysis was the UK National Health Service and social services.

**Setting** The intervention was provided in a community leisure centre setting.

**Participants** 86 young people aged 14–17 years attending Tier 2 and Tier 3 CAMHS (Child and Adolescent Mental Health Services) outpatient services presenting with depression.

**Interventions** The intervention comprised 12 separate sessions of circuit training over a 6-week period. Sessions were supervised by a qualified exercise therapist. Participants also received treatment as usual. The comparator group received treatment as usual.

**Results** We found improvements in the Children's Depression Inventory-2 (CDI-2) and estimated cost-effectiveness at £61 per point improvement in CDI-2 for the exercise group compared with control. We found no evidence that the exercise intervention led to differences in quality-adjusted life years (QALY). QALYs were estimated using the EQ-5D-5L (5-level version of EuroQol-5 dimension).

**Conclusions** There is evidence that exercise can be an effective intervention for adolescents with depression and the current study shows that preferred intensity exercise could also represent a cost-effective intervention in terms of the CDI-2.

**Trial registration number** NCT01474837.

## BACKGROUND

Depression is highly prevalent in adolescence[1] with the numbers of reported cases doubling between the mid-1980s and 2000s.[2] Among community samples of adolescents, the prevalence of major depressive disorder (MDD)

### Strengths and limitations of this study

► Economic analysis conducted alongside a randomised trial of an exercise intervention for adolescents with depression.
► This study uses costs and effectiveness data collected directly from the trial population to estimate the cost-effectiveness of an exercise intervention for adolescents with depression.
► Appropriate statistical methods were adopted to control for baseline characteristics and missing data.
► The small sample size at follow-up limits the strength of the conclusions made.

is reported to be between 4% and 8%.[3] Additionally, it is estimated that 12% of children and adolescents may have subthreshold symptoms of depression.[4] There is some evidence to suggest that exercise may have beneficial effects on depression in young people.[5–8] A recent feasibility randomised controlled trial compared a moderate to strenuous exercise intervention to stretching for adolescents diagnosed with MDD.[9] Statistically significant reductions in depression scores were observed for both groups. However, the study was limited by a small sample size (30 participants initially with 15 completing 12-month follow-up). For this reason, a randomised controlled trial was designed and conducted to determine the effectiveness of a preferred intensity exercise intervention on the depressive symptoms of adolescents with depression.[10 11]

With growing concern as to the affordability of healthcare, particularly in the context of the National Health Service (NHS) budget, it is important to consider the value for money afforded by any intervention in terms of both its

costs and effectiveness. Furthermore, the costs of child and adolescent mental health can be substantial with a recent study estimating total annual costs of emotional disorders in the UK at £1165 per person in 2007/2008 prices.[12] Individuals treated for depression during childhood can continue to incur substantial costs into adulthood.[13] These considerations should be contemplated when making resource allocation decisions.

However, there has been limited and inconclusive evidence looking at the health economic case for interventions for child or adolescent depression. One study we are aware of looked at an exercise-related intervention.[14] This was a cost-utility study of a dance intervention for adolescent girls with internalising problems. The dance intervention was considered cost-effective as it cost US$3830 per quality-adjusted life year (QALY). There have also been studies that conducted economic evaluations of adolescents with depression using non-exercise-based interventions. An economic evaluation was conducted in the USA alongside a randomised controlled trial for subjects with a major diagnosis of depression.[15 16] Groups received either placebo, fluoxetine, cognitive–behavioural therapy (CBT) or a combination of fluoxetine and CBT. Researchers found that combination therapy with fluoxetine and CBT was highly likely to be cost-effective at 36 weeks. A study of the cost-effectiveness of a collaborative care intervention was conducted on adolescents in the USA.[17] This found the intervention to be cost-effective, at $18 239 per QALY gained. There have also been three UK studies looking at CBT in adolescents with depression. Byford and colleagues evaluated CBT in addition to selective serotonin reuptake inhibitors (SSRI) with normal clinical care.[18] The comparator was SSRI plus normal clinical care alone. This study found only a 30% probability that CBT and SSRI would be more cost-effective than SSRI alone. Anderson and colleagues evaluated classroom-based CBT compared with usual school provision of Personal, Social and Health Education.[19 20] They found no evidence that the intervention was cost-effective. Computerised CBT was also compared with a website control in a feasibility study.[21] The authors concluded a future large-scale study was feasible but the study was not powered to show differences in effects.

In order to add to this literature and to aid in decision-making, we conducted an economic evaluation alongside the aforementioned clinical trial.[11] The aim of this economic evaluation was to examine the cost-effectiveness of a preferred exercise intervention in addition to treatment as usual compared with treatment as usual alone. This article describes the methods used to conduct this economic analysis. We present the results in terms of both costs and effects for the brief exercise intervention compared with treatment as usual.

## METHODS
### Randomised controlled trial
The current economic evaluation was conducted as an integral part of a randomised controlled trial.[11] In brief, this study compared a preferred intensity exercise intervention, in addition to treatment as usual, compared with a treatment as usual only control group. The sample was drawn from young people attending CAMHS (Child and Adolescent Mental Health Services) outpatient services in Nottingham City and Nottinghamshire County, UK. Participants would be attending either Tier 2 (typically CAMHS specialists working in primary and community care) and Tier 3 (typically multidisciplinary teams in a community mental health clinic providing a more specialised service). To be included in the study, participants needed to be: adolescents aged between 14 and 17; in receipt of treatment from a health or social care professional for depression; and scoring 14 or above on the Children's Depression Inventory-2 (CDI-2).[22] The study comprised 86 participants, who were individually randomised to study groups by means of sequentially numbered opaque sealed envelopes. The intervention package consisted of a maximum of 12 sessions, delivered over a 6-week period. Sessions consisted of aerobic exercise in the form of circuit training, tailored to the exercise preferences of participants. Exercise sessions were scheduled for 60 min and were given in groups with a maximum size of 10 participants. The sessions were preferred intensity as participants could choose the order in which they undertook different exercises, the intensity of their exercise and when to take breaks. Participants were followed up for 6 months. Informed written consent was obtained from the legal guardians of those under the age of 16, alongside the young person's assent. Informed written consent was obtained directly from those 16 years of age and over.

### Costs
The perspective was that of the NHS and social services. All costs were for the year 2012/2013, measured in UK pound sterling. As the time frame of the analyses was less than 1 year, neither costs nor outcomes were discounted. The attendance of participants in the treatment group was recorded at each session. Sessions were run by two members of research staff. As this service was run from a CAMHS unit, data on staff and non-staff overheads relevant to CAMHS were taken from a published source of healthcare unit cost data.[23] Actual costs were also recorded for other resource items required to provide the sessions, for example, the cost of room bookings.

Health and social care resource use data were collected using the client service receipt inventory (CSRI).[24] This is a comprehensive inventory of resource use that has been widely used in economic evaluations of mental health interventions and can be adapted to fit individual contexts. Its use allows resource use patterns to be described and these can then be costed using appropriate unit costs. This instrument is designed to be adaptable and has been used in a wide number of different diseases, settings and client groups. The CSRI was completed at baseline and at the 6-month follow-up period by means of a face-to-face interview conducted by a member of the study team who

**Table 1** Unit costs used in the analysis (UK£ 2012/2013)

| Resource use item | Cost | Source |
|---|---|---|
| Inpatient stays | Various | NHS reference costs[26] |
| Medicines | Various | British National Formulary[25] |
| Mental health A&E | 228 | NHS reference costs[26] |
| Accident and Emergency | 115 | NHS reference costs[26] |
| Mental health outpatient | 234 | NHS reference costs[26] |
| Other outpatient | 187 | NHS reference costs[26] |
| School nurse | 14 | PSSRU[23] |
| GP | 37 | PSSRU[23] |
| Paediatrician | 187 | NHS reference costs[26] |
| Physiotherapy | 12 | PSSRU[23] |
| Clinical psychology | 68 | PSSRU[23] |
| Speech therapy | 15 | PSSRU[23] |
| Hearing specialist | 65 | NHS reference costs[26] |
| Other contacts | 28 | Weighted average of reported contacts |
| Counselling/therapist | 59 | PSSRU[23] |
| Home help/care worker | 19 | PSSRU[23] |
| Social worker | 51 | PSSRU[23] |
| Overnight stay | 91 | PSSRU[23] |

A&E, Accident and Emergency; GP, general practitioner; NHS, National Health Service; PSSRU, Personal Social Services Research Unit.

was blinded to study group. The time frame used was the last 6 months for all questions. Data were collected on the following: medicines; inpatient stays; use of other hospital services; contacts with healthcare practitioners; and social care. The CSRI also covered resource use relating to overnight stays in children's homes and foster care alongside other services used by the respondent's family as a result of the young person's behavioural or mental health problems.

Resources identified were valued using the unit costs identified in table 1. Medicines were costed using the British National Formulary,[25] accessed in March 2014. Prices obtained were deflated to 2012/2013 using the consumer price index. Where it was unclear what medicine was prescribed, a practising general practitioner was consulted about typical prescribing for the reason given by the respondent. For inpatient stays, NHS reference costs were used.[26] Where a stay in a mental health facility was recorded, a cost per day of £611 was used. Where the reason for admission was clear, costs were based on appropriate NHS reference costs. Where the reason was not clear, we used weighted average costs. Accident and Emergency (A&E) and outpatient costs were taken from NHS reference costs.[26] For mental health-related A&E contacts, the cost used was that of the A&E Mental Health Liaison services. For other A&E visits, a weighted average of all non-admitted A&E contacts was used. Mental health

outpatient contacts were estimated using a weighted average of outpatient and community CAMHS contacts. For other outpatient visits we used the cost for a paediatrics outpatient visit.

A range of contacts with community healthcare professionals were recorded by the CSRI. The unit costs used for these are also shown in table 1. Where necessary, assumptions on duration of contact were made. The modified CSRI asked for contacts with a number of different types of counsellor. As no further details were available, we assumed that all counselling services had the same cost, taken from a published source.[23] The cost of a visit to a clinical psychologist was based on an assumed duration of contact of 30 min.[23] The cost of a social worker was obtained from a published source.[23] It was not clear what the duration of contacts would be so an assumption of 20 min was used. Individuals were asked for any overnight stays in the last 6 months in either a children's home, foster carer or any other residential placement. One individual reported a stay in the 'other residential placement' category, and only in the baseline period. In the absence of any other information, we assumed this was equivalent to foster care and used a cost of £91 per day.[23]

## Outcome measures

Two separate measures of outcome were used for the economic evaluations. First, we used the primary outcome measure from the clinical study, the Children's Depression Inventory (CDI-2)—a 28-item self-report questionnaire that assesses the severity of current/recent depressive symptoms in young people aged 7–17 over the preceding 2 weeks.[22] Questions on the CDI-2 have three possible responses: 0, corresponding to no symptoms; 1, corresponding to probably or mild symptoms; and 2, corresponding to definite or marked symptoms. This gives a range of scores from 0 to 56 with higher scores representing higher depressive symptom severity. A score of 14 and above is considered to indicate clinical levels of depression.[22] The CDI-2 was used in a cost-effectiveness study to estimate cost per point change in CDI-2. We also carried out a cost-utility study estimating the cost per QALY generated by the exercise intervention. QALYs were estimated using the EQ-5D-5L (5-level version of EuroQol-5 dimension) instrument.[27] This was scored using a published scoring algorithm.[28] Both outcome measures were administered at baseline, postintervention (approximately 6 weeks after commencement of exercise) and at the 6-month follow-up.

## Analysis

QALYs were estimated for the follow-up period using 'area under the curve'. To do this, we assumed a linear relationship between the three data collection points (baseline, postintervention and follow-up). QALYs were estimated using the actual time of each of the three data collection points so length of time could differ between respondents. All individuals who had follow-up resource use data had EQ-5D scores as these

two measures were taken together. However, there were four individuals for whom we had almost complete data except a single missing EQ-5D score in each case. The baseline EQ-5D score was imputed using mean value imputation for one individual.[29] Three individuals were missing EQ-5D at the 6-week follow-up and values were imputed using multiple imputation.[29] These four individuals, in addition to those individuals in who we had full health economics data, represented the complete case analysis. For individuals where the follow-up questionnaire was unavailable, costs, QALYs and difference in CDI-2 were imputed using baseline EQ-5D and CDI-2, time from randomisation to follow-up and baseline total costs. Multiple imputation was carried out in SPSS V.23 using 50 data sets (a 'rule of thumb' is the number of data sets should equal the percentage of missing data).[30]

Regression-based methods were used to allow for differences in baseline characteristics.[31] Differences between the intervention and the control group for both costs and outcomes were estimated using seemingly unrelated regression (using sureg command in STATA, V.11). Costs were estimated controlling for: study group, baseline EQ-5D-5L and baseline total cost. Estimates for differences in CDI-2 were controlled for: study group, baseline CDI-2 and time from baseline to final follow-up. Estimates for differences in QALY were controlled for: study group, baseline EQ-5D and time from baseline to follow-up. Estimates from the 50 imputed samples were combined using 'Rubin's Rules'.[32] To estimate uncertainty associated with estimates, we used bootstrap resampling with 250 replications drawn from each of the 50 imputed data sets, giving 12 500 replications in total. These were used to estimate cost-effectiveness acceptability curves (CEAC).[33]

## RESULTS

There were 86 individuals recruited to the study, 42 and 44 in the control and exercise groups, respectively. These individuals completed CSRI by interview at baseline. Of these, 42 (17 in control group and 25 in exercise group) completed the follow-up CSRI at interview.

Descriptive characteristics are given for all participants and for those in the complete case analysis (table 2). Baseline characteristics are similar between groups for both the full data set and the health economics complete case analysis. There were no statistically significant differences between the control and the intervention group for either the full data set or the complete health economics data.

## Costs

The total time required to provide exercise sessions, including set-up and travel, was 2 hours. The cost, including overheads, for two individuals for 2 hours was £129; additionally, there was a charge of £23 for use of space in the leisure centre. This gave an estimate of £152 for the cost of a group session. There were 44 individuals randomised to the intervention group. Participants joined one of seven different exercise groups and each group had 12 scheduled sessions. This gave 82 sessions (two sessions were cancelled due to low numbers) and hence a total estimated cost for the intervention of £12 464. Of those in the exercise group, eight did not attend any sessions. There were a total of 277 attendances by the other individuals. This gave an average cost per person per session of £45.

The NHS and social care costs incurred by participants are given in table 3 for the complete case analysis. Estimated average costs at baseline were £3312 and £3280 for control and intervention groups, respectively. These differences were not statistically significant (independent sample t-tests). Costs were much lower in the follow-up period at £1301 for the control group and £1889 for the intervention group (this includes £351 for the cost of the intervention). The difference in costs between groups was £589, with a 95% CI of –£507 to £1685.

## Outcomes

The outcome measures used in this study are shown in table 4. EQ-5D-5L scores increased between baseline and follow-up in both groups. It can also be seen that the CDI-2 scores decrease (improve) in each group. Estimated QALY was higher in the control group; however,

| Table 2 | Descriptive characteristics | | | |
|---|---|---|---|---|
| | **Full data set** | | **Complete case analysis** | |
| **Baseline characteristic** | **Control** | **Intervention** | **Control** | **Intervention** |
| Number | 42 | 44 | 17 | 25 |
| Percentage female (%) | 81 | 75 | 88 | 88 |
| Percentage white (%) | 98 | 95 | 100 | 92 |
| Age (years) | 15.4 | 15.4 | 15.5 | 15.3 |
| Baseline CDI-2 | 28.2 | 29.1 | 28.7 | 29.4 |
| Baseline EQ-5D | 0.82 | 0.78 | 0.82 | 0.74 |
| Time from recruitment to follow-up (weeks) | 39 | 37 | 36 | 36 |

CDI-2, Children's Depression Inventory-2; EQ-5D, EuroQol-5 dimension.

**Table 3** Costs at baseline and in the follow-up period for control and intervention groups (£'s)

| | Baseline | | Follow-up | |
| --- | --- | --- | --- | --- |
| Resource use | Control (mean (SD)) | Intervention (mean (SD)) | Control (mean (SD)) | Intervention (mean (SD)) |
| Medicines | 8.5 (22.7) | 28.8 (121.1) | 4.3 (11) | 9.5 (21) |
| Inpatient stays | 1142 (4203) | 1046 (4982) | 0 (0) | 44.6 (162.9) |
| Mental health-related Accident and Emergency | 40.2 (89.6) | 36.5 (85.3) | 40.2 (89.6) | 73 (205.2) |
| Other Accident and Emergency | 47.4 (81.9) | 27.6 (60.1) | 20.3 (45.2) | 13.8 (50.6) |
| Mental health outpatient appointments | 1432 (1892) | 983 (1317) | 771 (1281) | 983 (1525) |
| Other outpatient | 44 (140.7) | 29.9 (88.4) | 110 (453.5) | 89.8 (448.8) |
| **Total costs of secondary care** | **2705 (5775)** | **2122 (5570)** | **941 (1310)** | **1204 (1632)** |
| School nurse | 11.3 (21.8) | 71.7 (272) | 22.5 (67.1) | 0 (0) |
| Health visitors | 0 (0) | 0 (0) | 0 (0) | 0 (0) |
| Cost of dental treatment | 10.3 (13.4) | 13.3 (13.9) | 13.9 (18.6) | 8.2 (14.5) |
| Cost of GP visits | 126.2 (88.8) | 126 (175) | 132.8 (107.9) | 72.5 (91.6) |
| Cost of visits to paediatrician | 44 (105.1) | 37.4 (93.5) | 0 (0) | 15 (74.8) |
| Optician visits | 5.9 (9.4) | 8 (12.9) | 1.2 (4.9) | 7.2 (24.4) |
| Physiotherapist | 2.8 (11.6) | 3.8 (16.9) | 0 (0) | 0 (0) |
| Clinical psychology | 0 (0) | 3.6 (18) | 127.1 (358.7) | 0 (0) |
| Speech therapy | 0.9 (3.6) | 0 (0) | 0 (0) | 0 (0) |
| Hearing specialist | 0 (0) | 18.2 (69) | 0 (0) | 2.6 (13) |
| Other visits to health practitioners | 37.5 (59.6) | 13.9 (39.4) | 0 (0) | 0 (0) |
| **Total costs of visits to health practitioners** | **239 (140)** | **296 (358)** | **297.5 (428.3)** | **105.5 (117.9)** |
| Family counselling | 31.2 (88.8) | 2.4 (11.8) | 0 (0) | 0 (0) |
| Individual counselling | 226 (282) | 316 (423) | 38.2 (142.9) | 125.1 (589.6) |
| Other counselling | 76.4 (138) | 14.2 (49) | 13.9 (57.2) | 0 (0) |
| **Total costs of counselling** | **333 (375)** | **333 (414)** | **52.1 (150.3)** | **125 (590)** |
| Home help/care worker | 0 (0) | 6.8 (34.2) | 0 (0) | 15.2 (76) |
| Social worker | 6 (24.7) | 61.2 (250) | 0 (0) | 40.8 (204) |
| After school clubs | 0 (0) | 0 (0) | 0 (0) | 0 (0) |
| Other support services | 1.1 (4.6) | 0 (0) | 0 (0) | 0 (0) |
| **Total costs of social care** | **7.1 (24.9)** | **68 (250)** | **0 (0)** | **56 (280)** |
| Nights in children's home | 0 (0) | 0 (0) | 0 (0) | 0 (0) |
| Nights in foster care | 0 (0) | 0 (0) | 0 (0) | 0 (0) |
| Nights in other residential care | 0 (0) | 364 (1820) | 0 (0) | 0 (0) |
| **Total costs for residential care** | **0 (0)** | **364 (1820)** | **0 (0)** | **0 (0)** |
| Service use by family related to respondents mental health | 19.9 (64.1) | 67.6 (137) | 5.6 (16.2) | 38.4 (139.8) |
| Costs of intervention | – | – | 0 (0) | 351 (144) |
| **Total costs of follow-up** | **3312 (5980)** | **3280 (6236)** | **1301 (1512)** | **1889 (1853)** |

Based on a sample of 42 cases (17 in control group and 25 in intervention group).
GP, general practitioner.

this group has a higher starting EQ-5D-5L value, so mean unadjusted QALYs would favour this group. Therefore, regression methods were used to estimate both QALY and CDI-2 differences between the exercise and the control group. These results can be seen in table 5. The estimated difference in QALY score was −0.0027 for the complete case analysis and 0.0019 for the imputed analysis, both with wide CIs. Effects in terms of CDI-2 were slightly larger for the imputed analysis, −4.8 compared with −4.6 for the complete case analsyis. These indicate that improvements in CDI-2 score were greater in the exercise group compared with the control group.

Estimates of cost-effectiveness are also shown in table 5, in terms of incremental cost-effectiveness ratios (ICER). These show the additional cost of generating each additional unit of effect. To be consistent with the cost per QALY analysis, we have changed the sign of the difference in CDI-2 scores when calculating the ICER. This is because a decrease in

**Table 4** Outcome measures used in the economic analyses at different study time points

| Outcome measure | Control mean (95% CI) | Intervention mean (95% CI) |
|---|---|---|
| Baseline EQ-5D-5L | 0.818 (0.741 to 0.895) | 0.744 (0.683 to 0.804) |
| Postintervention EQ-5D-5L | 0.841 (0.788 to 0.893) | 0.823 (0.761 to 0.885) |
| Follow-up EQ-5D-5L | 0.881 (0.836 to 0.926) | 0.838 (0.768 to 0.908) |
| Time in study (weeks) | 36.300 (31.4 to 41.3) | 36.300 (32.6 to 40) |
| QALY | 0.594 (0.507 to 0.681) | 0.567 (0.494 to 0.641) |
| Baseline CDI-2 | 28.7 (25.1 to 32.4) | 29.4 (25.2 to 33.6) |
| Follow-up CDI-2 | 24.6 (19.3 to 29.8) | 20.4 (15.9 to 24.8) |

CDI-2, Children's Depression Inventory-2; EQ-5D-5L, 5-level version of EuroQol-5 dimension; QALY, quality-adjusted life year.

the CDI-2 represents an improvement in symptoms so is a positive benefit. Because of the small numbers for whom we have data, the results are subject to considerable uncertainty and should be treated with caution. The cost per point improvement in the CDI-2 was £73. This decreased to £61 for the imputed analysis. The complete case analysis showed a small decrease in QALYs. This meant the ICERs were negative, though with considerable uncertainty. The imputed analysis showed very small positive QALY gains and an ICER of £152 822, again with considerable uncertainty. CEACs for these results are shown in figures 1 and 2 for the imputed analysis. Figure 1 indicates the probability that the intervention is cost-effective at different values of a point improvement in the CDI-2. There is a 50% probability that the intervention is cost-effective at values of a point change in CDI-2 of approximately £65. Due to considerable uncertainty and the very small incremental effect on QALYs, the CEAC for the cost/QALY analysis shows much lower probability that the intervention is cost-effective at around 33%–37% at a cost per QALY of £20 000–£30 000 (figure 2).

## DISCUSSION

We found that the intervention leads to improvements in the CDI-2 score, costing £73 and £61 per point improvement in CDI-2 for the complete case and imputed data, respectively. Mean costs in the follow-up period were higher for the intervention group, mainly due to the cost of the intervention. We found no evidence that the intervention was associated with improvements in QALYs. This study comprised the largest trial of its type in a clinical population

of adolescents with depression. Additionally, as the clinical study had a parallel economic evaluation, we were able to directly assess the costs and cost-effectiveness of this intervention. Data on resource use were collected in face-to-face interviews ensuring that that completion of individual questions was high for those who completed follow-up. Additionally, outcome measures and resource use questions were collected in the same interview ensuring that where we had resource use data we also had EQ-5D-5L data.

One weakness of the current study was that there was considerable loss to follow-up at the final time point. This was less of an issue for the study primary outcome measure, which was based on CDI-2 at the 6-week follow-up, but was more problematic for the economic evaluation. Only those who completed the final assessment would have completed the follow-up CSRI, so only 49% of participants could be included in the complete case analysis. We used multiple imputation to estimate costs and QALYs for missing cases. This produced a lower estimate of the cost per point change in CDI-2. Another potential disadvantage was that there were often delays between baseline data collection and starting the intervention. In the control group, there were similar delays. This meant that the total length of follow-up varied between participants. Mean follow-up was 36 weeks in both groups. As the period of recall was 26 weeks for the CSRI, this did not cover all costs between baseline and follow-up in all participants. This may have understated total costs in the follow-up period as average 6-month costs are declining over time.

The study estimated that differences between baseline and follow-up CDI-2 were greater in the intervention

**Table 5** Results of the economic analysis for both cost per point change in CDI-2 and cost per QALY

| Outcome | Analysis | Incremental cost | CI | Incremental effect | CI | ICER |
|---|---|---|---|---|---|---|
| CDI-2 difference | Complete case | 334 | (−606, 1274) | −4.6 | (−10.1, 0.87) | 73* |
| | Imputed | 292 | (−558, 1142) | −4.8 | (−10.3, 0.75) | 61* |
| QALY difference | Complete case | 355 | (−596, 1305) | −0.0027 | (−0.04, 0.03) | Negative |
| | Imputed | 286 | (−1063, 1634) | 0.0019 | (−0.063, 0.067) | £152 822 |

*For ICERs related to change in CDI-2, the sign of the difference has been changed as a negative change in CDI-2 represents an improvement.
CDI-2, Children's Depression Inventory-2; ICER, incremental cost-effectiveness ratio; QALY, quality-adjusted life year.

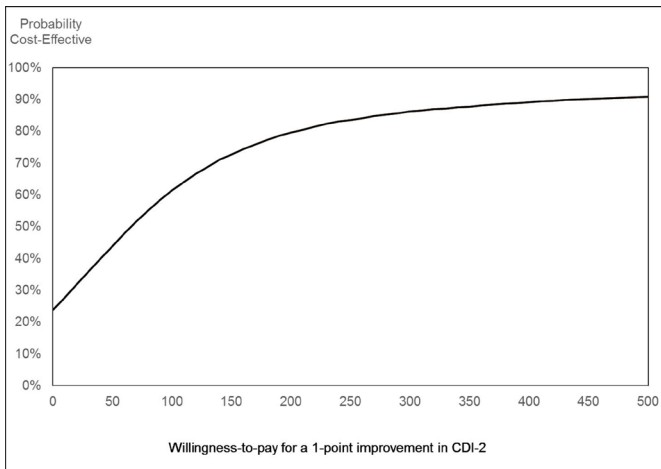

**Figure 1** Cost-effectiveness acceptability curve for cost per point improvement in Children's Depression Inventory-2 (CDI-2).

group. We found no evidence to suggest that there were any differences in QALYs generated between groups. A number of different methods have been used to estimate QALYs. A US study used the Health Utilities Index (HUI3) in an economic evaluation of a dance intervention for adolescent girls with internalising problems.[14] This intervention was considered cost-effective (US$3830 per QALY); however, there were differences in how the QALY effect was calculated, that is, this used QALY differences from baseline utility values whereas our study used regression-based methods to allow for differences in baseline characteristics. Two other US studies used depression-specific outcome measures to define an individual's level of depression in order to combine this with estimates of QALY effects of depression level. A US study estimated QALYs by means of depression-free days.[15 16] QALYs were estimated by assigning a value of 1.0 for depression-free days, and a value of 0.6 for days with depression. Wright *et al* used a similar approach where levels of depression were defined,

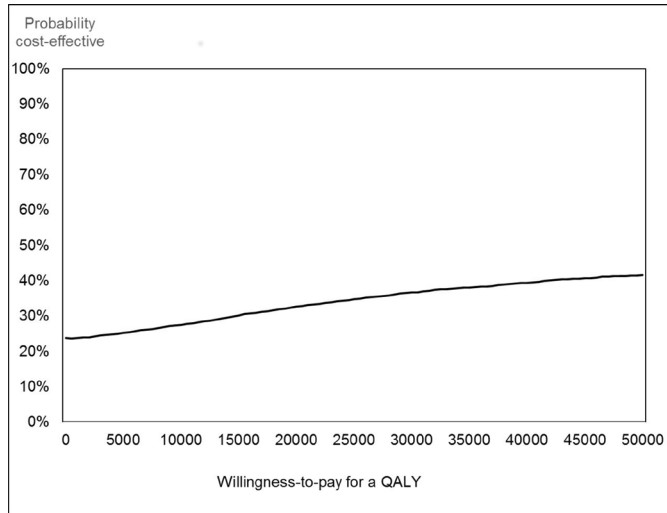

**Figure 2** Cost-effectiveness acceptability curve for cost per quality-adjusted life year (QALY) analysis.

again these were valued by utility weights for these states.[17] Both these approaches suggested differences in utility from the interventions. We are aware of four published studies, reporting on three different trials, that evaluated QALYs in adolescents with depression using the EQ-5D-3L.[18–21] None of these studies found that the intervention led to improvements in QALYs and did not find the intervention to be cost-effective.

The fact that the main clinical paper[11] and the current economic evaluation indicated differences in CDI-2 scores but not in the EQ-5D-5L or QALYs may therefore be due to a number of factors. First, the EQ-5D may be unsuitable for use in adolescents with depression, that is, it may be insensitive to changes that can be detected by depression-specific measures. We found only one published study that evaluated the performance of the EQ-5D in adolescents with depression.[34] This study found a statistically significant correlation between EQ-5D and depression-specific outcome measures, though this relationship was stated to be weak. However, there is considerable evidence looking at the use of EQ-5D in adults with depression. A recent systematic review found 14 studies which evaluated the performance of the EQ-5D in patients with depression and/or anxiety.[35] The mean age of participants in these studies ranged from 39 to 49 years. The authors of this systematic review concluded that the EQ-5D showed good construct validity and responsiveness for people with depression. These studies provide limited evidence for the use of EQ-5D in adolescents with depression.

Another factor may be that the current study used the EQ-5D-5L.[27] All the currently available literature evaluating the EQ-5D in depression uses the EQ-5D-3L.[36] It may be that the 5-level version performs less well in people with depression. An alternative explanation for the small non-significant estimate of the QALY effect of the intervention may simply have been that the sample size was insufficient to detect differences in this generic measure, which we would expect to be less sensitive than the depression-specific score. Alternatively, there may be differences in depression-related symptoms but these may not be translated into improvements in overall health-related quality of life. Further research using and evaluating the EQ-5D-5L in adolescents would be beneficial.

The current study suggests that the intervention can generate improvements in the CDI-2 at approximately £61 per point change in CDI-2 (imputed analysis). This may represent a cost-effective intervention but it is unclear as to what a point change in the CDI-2 should be worth. A study in Korean adolescents suggested that cut of points for mild, moderate and severe depression on the CDI-2 could be 15, 20 and 25, respectively.[37] A recent meta-analysis of utility values of health states related to depression in adults suggested that mild, moderate and severe depression could be associated with EQ-5D utility values of 0.57, 0.52 and 0.39, respectively.[38] This implies that changes in depression states can have significant implications for EQ-5D, and hence QALYs, and provides some evidence that the CDI-2 differences shown in this study could be meaningful.

Additionally, it is unclear how persistent any benefits generated by the intervention would be. The current study only followed participants up for approximately 6 months. If benefits persisted for more than 6 months they would not be captured by the current design, which therefore may overestimate the cost per point change in CDI-2. Furthermore, the delivery of the intervention in this study was influenced because it was part of a clinical trial. This limited the numbers available for each exercise group as some of the potentially eligible participants would have been randomised into the control group. In practice, the intervention could be delivered to slightly larger groups with the same resources and hence costs per person may be lower than indicated in this trial.

## CONCLUSION

Mental health practitioners are committed to using the best available evidence to guide their practice. It is important to demonstrate that any new intervention is cost-effective as well as being clinically effective. There is evidence that exercise can be an effective intervention for adolescents with depression[11] and the current study shows that preferred intensity exercise could also represent a cost-effective intervention in terms of the CDI-2. However, more work would be needed to establish the health economic value of a point change in CDI-2. Incorporating exercise into the repertoire of interventions could add value to the care of children and adolescents receiving treatment and support for depression.

**Acknowledgements** The authors acknowledge the contribution of all study participants and their parents, and staff of the Child and Adolescent Mental Health Services at Nottinghamshire Healthcare NHS Trust, without whom this study would not have been possible.

**Contributors** PC and TC lead the clinical trial. BG was the trial statistician. TS designed the original economic analysis. DT amended data collection forms and conducted the economic analysis. DT drafted the manuscript. All authors commented on the manuscript and input into relevant sections. All authors read and approved the final manuscript. DT takes responsibility for integrity of the health economics analysis reported in the study.

**Funding** The study was funded by the National Institute for Health Research, Research for Patient Benefit Programme (grant reference number: PB-PG-1208-18097).

**Disclaimer** The funders played no role in the collection, analysis and interpretation of data, or in the writing of this manuscript.

**Competing interests** None declared.

**Ethics approval** Ethical approval was received from Nottingham Research Ethics Committee on 18 July 2011(REC reference: 11/EM/).

**Provenance and peer review** Not commissioned; externally peer reviewed.

**Data sharing statement** No additional data are available.

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
