## [Reviewer comments · BMJ Open]

ARTICLE DETAILS

TITLE (PROVISIONAL)	Cost-effectiveness of a preferred intensity exercise programme for young people with depression compared to standard care. An economic evaluation alongside a clinical trial in the UK
AUTHORS	Turner, David; Carter, Timothy; Sach, Tracey; Guo, Boliang; Callaghan, Patrick

VERSION 1 – REVIEW

REVIEWER	Leena Forma Faculty of Social Sciences (health sciences) and Gerontology Research Center (GEREC) University of Tampere Finland
REVIEW RETURNED	27-Mar-2017

GENERAL COMMENTS	The manuscript reports the results of cost-effectiveness and cost-utility analyses based on a RCT. Depression among adolescents is a major public health problem and therefore this study is well motivated. The analyses have been performed correctly and the paper is well written. My comments are mainly related to the clarity of reporting. Major comments: 1. Description of the intervention. Although there is a study protocol published, the main information should be stated in this manuscript, too. The intervention is a preferred intensity exercise programme, but the “preferred” remains unclear.2. Setting. For me living outside of the UK Tier 2 and Tier 3 CAMHS are not known. Please give a short description.3. Abstract. It would be good to mention EQ-5D-5L already in the abstract.4. Measuring costs. Client service receipt inventory (CSRI) could be described more exactly. It is first described as a face-to-face interview, but on page 9 it is referred to as a completed questionnaire.5. Abbreviations. Some abbreviations may be established but were not familiar to me: A&E, O/P. These should be opened up.6. Analysis. All statistical testing are not exactly described. On page 10, line 7, which was the test?7. Table 4. The title is not clear, for there is also CDI-2 reported in the table.8. Table 5. Title is missing. In calculating ICER you changed the sign of the difference in CDI-2. Please add this information also into a footnote of the table.9. Please consider providing CEA planes.10. Figure 1 and 2. Please insert axis titles.
--

	Minor comments: 1. Page 10, line 28, did not the cancelled sessions cause any costs? 2. Table 3. Please correct the title (CostsCosts)
--	--

REVIEWER	Jesper Krogh Department of Endocrinology Rigshospitalet, Copenhagen University Hospital, Copenhagen, Denmark I have published a number of studies on exercise and depression with negative results.
REVIEW RETURNED	11-Apr-2017

GENERAL COMMENTS	A preferred intensity exercise programme for young people with depression: a cost-effectiveness study. By David Turner et al. This is a sub-analysis of a previously published trial. My main concerns are the lack of a pre-defined relevant cost estimate, e.g., how would the authors define a relevant cost for a CDI point. My other concern relates to the analysis. There were a substantial amount of missing data and it was somewhat unclear how this was handled. Regardless of approach the number of missing data was substantial questioning the relevance of data. However, as I understand from the manuscript, the lack of other publically available data may justify publication. Abstract By reading the abstract it is no clear to me, whether this is a sub-group analysis of a previously conduted RCT or publication of secondary outcomes? What does 'A 'within trial' cost-effectiveness and cost-utility analysis conducted alongside a randomised controlled trial' mean? Isn't this just a analysis of secondary endpoints from a RCT? If this is results from a RCT please report according to CONSORT, including changing the title. What are the outcomes How was participants allocated to either intervention or TAU? I realize there may be some problems with duplications with previous reporting, but we need to be presented with the change in depression scores. I do not think it is enough to present a cost-estimate to be able to assess whether this is a meaningful cost. The conclusion should be rewritten. It states that this is an effective intervention, but the outcome relates to cost-effectiveness. Methods Was the trial registered at clinicaltrials.gov or similar? Is there a published protocol? Please describe randomization procedures in detail Please describe who was blinded – if any Costs should be under the Outcomes heading Please specify what was considered a 'reasonable' cost for 1 CDI point Please specify, when performing imputation, which variables were used as predictors? I would prefer that the main analysis would be based on the imputed data and not on complete case analysis since this is known to bias the results. Please specify what 'Regression based methods' are. Please specify what a clinical relevant effect on CDI is?
--

VERSION 1 – AUTHOR RESPONSE

Reviewer 1

Major comments:

Comment 1: Description of the intervention. Although there is a study protocol published, the main information should be stated in this manuscript, too. The intervention is a preferred intensity exercise programme, but the “preferred” remains unclear.

Response: We have included the following text to explain the preferred intensity aspect: “The sessions were preferred intensity as participants could choose the order in which they undertook different exercises, the intensity of their exercise, and when to take breaks.” Page 6

Comment 2: Setting. For me living outside of the UK Tier 2 and Tier 3 CAMHS are not known. Please give a short description.

Response: We have added following description. Participants would be attending either tier 2 (typically CAMHS specialists working in primary and community care) and tier 3 (typically multidisciplinary teams in a community mental health clinic providing a more specialised service). Page 6.

Comment 3: Abstract. It would be good to mention EQ-5D-5L already in the abstract.

Response: Have now done so.

Comment 4: Measuring costs. Client service receipt inventory (CSRI) could be described more exactly. It is first described as a face-to-face interview, but on page 9 it is referred to as a completed questionnaire.

Response: Have added the following text (page 7): “This is a comprehensive inventory of resource use that has been widely used in economic evaluations of mental health interventions and can be adapted to fit individual contexts. Its use allows resource use patterns to be described and these can then be costed using appropriate unit costs. This instrument is designed to be adaptable and has been used in a wide number of different diseases, settings and client groups.” Have also amended the section specified to make clear that this was completed at interview.

Comment 5: Abbreviations. Some abbreviations may be established but were not familiar to me: A&E, O/P. These should be opened up.

Response: Have defined A&E (accident and emergency) at first use. Have replaced O/P with outpatient in all cases.

Comment 6. Analysis. All statistical testing are not exactly described. On page 10, line 7, which was the test?

Response: T-Tests – have included this in relevant section. Have also added more detail in analysis section.

Comment 7. Table 4. The title is not clear, for there is also CDI-2 reported in the table.

Response: Have amended with following. "Outcome measures used in the economic analyses at different study time points"

Comment 8. Table 5. Title is missing. In calculating ICER you changed the sign of the difference in CDI-2. Please add this information also into a footnote of the table.

Response: Apologies for missing this title. Have added the following: "Table 5 – Results of the economic analysis for both cost per point change in CDI-2 and cost per QALY" Have also added a footnote to the table to explain the change in sign.

Comment 9. Please consider providing CEA planes.

Response: Could supply these if requested by editors, either in text or as supplementary material. However, we don't consider they add anything in addition to the CEACs so have not currently included these.

Comment 10. Figure 1 and 2. Please insert axis titles.

Response: Have added axis titles.

Minor comments:

Comment 1. Page 10, line 28, did not the cancelled sessions cause any costs?

Response: No, it was assumed that costs were accrued for only the sessions that occurred. Staff could have done other activities in this time, and hence the opportunity cost was not necessarily incurred. Room booking fees may still have been incurred but this would only have made a difference of £0.16 per attendance.

Comment 2. Table 3. Please correct the title (CostsCosts)

Response: Thank you for pointing out this mistake, have amended.

Reviewer 2

Comment: This is a sub-analysis of a previously published trial. My main concerns are the lack of a pre-defined relevant cost estimate, e.g., how would the authors define a relevant cost for a CDI point.

Response: We acknowledge this is an issue. Cost-effectiveness analyses, such as the cost per point change in CDI-2 reported here, can be useful and use measures that are likely to be sensitive to change in diseases. However, they also have problems with interpretation compared to cost per QALY where there are more accepted values for willingness to pay for a QALY. Have cited evidence to show CDI-2 values for mild moderate and severe depression cut-of points and also possible utility values associated with these depression states (discussion, page 17). This contextualises the results presented.

Comment: My other concern relates to the analysis. There were a substantial amount of missing data and it was somewhat unclear how this was handled. Regardless of approach the number of missing data was substantial questioning the relevance of data. However, as I understand from the manuscript, the lack of other publically available data may justify publication.

Response: We agree that missing data was a problem. Have clarified further how imputation was carried out. Also, have amended analysis to allow for more imputation data sets as recommended in the literature (i.e., 1 per each 1 % of missing data – ref).

Comment: Abstract

By reading the abstract it is no clear to me, whether this is a sub-group analysis of a previously conducted RCT or publication of secondary outcomes?

Response: This work describes the economic evaluation carried out alongside the clinical trial. It was always intended that this work would be carried out alongside the trial and this was specified in the published protocol paper (Carter et al 2012). So this is not a sub-group analysis though could possibly be described as a publication of secondary outcomes.

Comment: What does 'A 'within trial' cost-effectiveness and cost-utility analysis conducted alongside a randomised controlled trial' mean? Isn't this just a analysis of secondary endpoints from a RCT?

Response: It is quite common to conduct an economic analysis alongside a clinical trial, the issues relating to this have been discussed by Petrou and Gray, 2011. ¹ In order to have an economic analysis as part of a clinical trial it has to be specifically designed into the RCT. So we would not describe this as an analysis of secondary endpoints, rather it is reporting an analysis that was always intended as part of the study. We have removed the language relating to 'within' trial cost-effectiveness to avoid confusion.

Comment: If this is results from a RCT please report according to CONSORT, including changing the title.

Response: The result is from an economic evaluation conducted alongside a clinical trial. The clinical trial has been reported elsewhere, according to consort guidelines. However, the BMJ open specifies that economic evaluations follow the CHEERS guidance, which includes guidance for economic evaluation alongside clinical trials. The CHEERS statement was completed and accompanied the original submission.

Comment: How was participants allocated to either intervention or TAU?

Response: Page 6 of the health economics paper has the following: "The study comprised 86 participants, who were individually randomised to study groups by means of sequentially numbered opaque sealed envelopes."

The main clinical data gives more detail: "The random allocation sequence was computer generated by the trial statistician using permuted block randomisation with varying block size. In order to ensure allocation concealment sequentially numbered opaque sealed envelopes were used. Individuals were randomised to groups by a researcher not connected to the study team."²

Comment: I realize there may be some problems with duplications with previous reporting, but we need to be presented with the change in depression scores. I do not think it is enough to present a cost-estimate to be able to assess whether this is a meaningful cost.

The estimated incremental effect in terms of change in CDI-2 scores for the intervention group compared to the control group are given in Table 5. These are -4.6 and -4.8 for the complete case and imputed analysis respectively. Results in terms of cost per point improvement in CDI-2 are also presented.

The conclusion should be rewritten. It states that this is an effective intervention, but the outcome relates to cost-effectiveness.

Response: Have modified conclusion

Comment: Methods

Was the trial registered at clinicaltrials.gov or similar?

Response: Yes.

Comment: Is there a published protocol?

Response: Yes, have added this in as a reference.

Comment; Please describe randomization procedures in detail

Response: Please see earlier point

Comment; Please describe who was blinded – if any

Response: Outcome assessors, including the data input administrator, were blind to treatment group at both follow up time points. This is stated in the main trial paper.

Comment: Costs should be under the Outcomes heading

Response: Costs and outcomes are both sub-headings in the methods sections. These are both important aspects of an economic analysis so they have been given separate sections.

Comment: Please specify what was considered a 'reasonable' cost for 1 CDI point

Response: We are not aware of any literature that establishes what should be a reasonable amount that a decision maker should pay for a point improvement in CDI. However, we are aware of literature that relates CDI-2 scores to depression severity states. And there is also literature that relates depression severity states to EQ5D scores. This gives an estimate of the potential QALY impact of a 5 point change in CDI-2 scores, which was the order of effect seen in this trial. This has been added to the discussion section to help contextualise these results.

Comment: Please specify, when performing imputation, which variables were used as predictors?

Response: Have added a section to the 'analysis' section offering further detail on this.

I would prefer that the main analysis would be based on the imputed data and not on complete case analysis since this is known to bias the results.

We have changed the emphasis to give more weight to imputed analysis.

Comment: Please specify what 'Regression based methods' are.

Response: Seemingly unrelated regression using the `sureg` command in STATA version 11. This is stated on page 9.

Comment; Please specify what a clinical relevant effect on CDI is?

Responses: See previous comment.

1. Petrou S, Gray A. Economic evaluation alongside randomised controlled trials: design, conduct, analysis, and reporting. . BMJ 2011;342(d1548).

2. Carter T, Guo B, Turner D, et al. Preferred intensity exercise for adolescents receiving treatment for depression: a pragmatic randomised controlled trial BMC Psychiatry 2015;15(1):247.

VERSION 2 – REVIEW

REVIEWER	Leena Forma Faculty of Social Sciences (health sciences) University of Tampere Finland
REVIEW RETURNED	18-Aug-2017
GENERAL COMMENTS	The authors have revised the manuscript sufficiently.